# Clinical Relevance and Interplay between miRNAs in Influencing Glioblastoma Multiforme Prognosis

**DOI:** 10.3390/cells13030276

**Published:** 2024-02-02

**Authors:** Samantha Epistolio, Giulia Dazio, Ismail Zaed, Nora Sahnane, Debora Cipriani, Francesco Polinelli, Jessica Barizzi, Paolo Spina, Federico Mattia Stefanini, Michele Cerati, Sergio Balbi, Luca Mazzucchelli, Fausto Sessa, Gianfranco Angelo Pesce, Michael Reinert, Andrea Cardia, Francesco Marchi, Milo Frattini

**Affiliations:** 1Laboratory of Genetics and Molecular Pathology, Institute of Pathology, Ente Ospedaliero Cantonale (EOC), 6900 Locarno, Switzerland; samantha.epistolio@eoc.ch (S.E.); giuliaramelli23@hotmail.com (G.D.); jessica.barizzi@eoc.ch (J.B.); paolo.spina@eoc.ch (P.S.); luca.mazzucchelli@eoc.ch (L.M.); 2Service of Neurosurgery, Neurocenter of the Southern Switzerland, Regional Hospital of Lugano, Ente Ospedaliero Cantonale (EOC), 6900 Lugano, Switzerland; ismaiel.zaed@eoc.ch (I.Z.); debora.cipriani@ksa.ch (D.C.); francesco.polinelli@eoc.ch (F.P.); michael.reinert@hin.ch (M.R.); andrea.cardia@eoc.ch (A.C.); francesco.marchi@eoc.ch (F.M.); 3Unit of Pathology, Department of Medicine and Technological Innovation, University of Insubria, ASST Sette Laghi, 21100 Varese, Italy; nora.sahnane@asst-settelaghi.it (N.S.); michele.cerati@asst-settelaghi.it (M.C.); fausto.sessa@uninsubria.it (F.S.); 4Department of Environmental Science and Policy, Faculty of Science and Technology-ESP, University of Milan, 20122 Milan, Italy; federico.stefanini@unimi.it; 5Division of Neurological Surgery, Department of Biotechnology and Life Sciences, University of Insubria, ASST Sette Laghi, 21100 Varese, Italy; sergio.balbi@uninsubria.it; 6Faculty of Biomedical Sciences, Università della Svizzera italiana, 6900 Lugano, Switzerland; 7Radiation Oncology, Oncology Institute of Southern Switzerland, Ente Ospedaliero Cantonale (EOC), 6501 Bellinzona, Switzerland; gianfrancoangelo.pesce@eoc.ch; 8Faculty of Medicine, University of the Southern Switzerland, 6900 Lugano, Switzerland

**Keywords:** glioblastoma, miRNAs, miRNA pairs, overall survival, clinical outcome, temozolomide

## Abstract

Glioblastoma multiforme (GBM) is usually treated with surgery followed by adjuvant partial radiotherapy combined with temozolomide (TMZ) chemotherapy. Recent studies demonstrated a better survival and good response to TMZ in methylguanine-DNA methyltransferase (*MGMT*)-methylated GBM cases. However, approximately 20% of patients with *MGMT*-unmethylated GBM display an unexpectedly favorable outcome. Therefore, additional mechanisms related to the TMZ response need to be investigated. As such, we decided to investigate the clinical relevance of six miRNAs involved in brain tumorigenesis (miR-181c, miR-181d, miR-21, miR-195, miR-196b, miR-648) as additional markers of response and survival in patients receiving TMZ for GBM. We evaluated miRNA expression and the interplay between miRNAs in 112 *IDH* wt GBMs by applying commercial assays. Then, we correlated the miRNA expression with patients’ clinical outcomes. Upon bivariate analyses, we found a significant association between the expression levels of the miRNAs analyzed, but, more interestingly, the OS curves show that the combination of low miR-648 and miR-181c or miR-181d expressions is associated with a worse prognosis than cases with other low-expression miRNA pairs. To conclude, we found how specific miRNA pairs can influence survival in GBM cases treated with TMZ.

## 1. Introduction

Among malignant tumors arising in the brain, glioblastoma multiforme (GBM) is the most diffuse, characterized by a median overall survival (OS) of 12–15 months from the time of diagnosis; therefore, among neoplastic diseases, it has one of the worst prognoses [1,2]. This behavior mainly depends on the infiltrating growth nature and abundant vascularization; these factors lead to a rapid progression of the disease. The most appropriate treatment for gliomas used to be safe, optimal surgical resection followed by adjuvant chemoradiation, represented by brain radiotherapy combined with temozolomide (TMZ) [3]. Recent studies have shown an improved OS for GBM treated with gross total resection (GTR) [3,4,5,6]. In addition, the literature describes how the survival probability depends on the response to chemotherapy, which is associated with the presence of *methylguanine-DNA methyltransferase* (*MGMT*) promoter methylation. The *MGMT* methylation pattern has the main role in the management of patients affected by GBM: *MGMT* promoter hypermethylation causes the abolishment of MGMT protein expression and, as a consequence, favors a better response to temozolomide (TMZ), leading to a significant improvement in patient outcomes [3,7,8]. However, a non-negligible portion of the patients affected by GBM and with an absence of *MGMT* methylation (about 20%) experience an unexpectedly favorable outcome after chemoradiation. Thus, additional mechanisms must be related to the response to TMZ [9,10,11]. A possible explanation could be related to a methylation-independent mechanism underlying MGMT expression regulation, in which micro-RNAs (miRNAs) may play a pivotal role. Micro-RNAs are a particular subgroup of noncoding RNAs with regulatory functions and a length between 18 and 25 nucleotides (nt). After post-transcriptional changes, miRNAs appear in the cytosol of the cell as single-stranded regulatory molecules. In combination with several miRNA-target proteins, they form RNA-inducing silencing complexes (miRISCs), which inhibit the translation of mRNAs due to the presence of the complementary sequence of nucleotides in the miRNAs themselves [12]. A number of studies have focused on single miRNAs and their influence on GBM survival. GBM-specific miRNAs can be both oncogenes and tumor suppressor genes, may lead to an absence of chemoradiotherapy efficacy, improve neo-angiogenesis and cell duplication, and regulate apoptosis [13,14,15,16,17,18]. Recently, our research group described how the expression of MGMT, evaluated via immunohistochemistry (IHC), is significantly associated with the expression of miR-181c, miR-195, and miR-648 [19]. In addition, we reported how *MGMT*-unmethylated cases are associated with low levels of miR-181d and miR-648 and how methylated GBM cases are associated with a low expression of miR-196b. Regarding survival, we observed a better OS in the absence of the MGMT protein using IHC, in *MGMT-*methylated patients, and in the cases of high miR-21 or miR-196b expression [19]. In addition, a better progression-free survival (PFS) was associated with the presence of *MGMT* promoter methylation and GTR but not with immunohistochemical MGMT protein expression and miRNA expression [19]. These data on individual miRNAs are relevant but not exhaustive for describing the miRNA-related process of response and survival in patients receiving TMZ. Indeed, the regulatory mechanisms mediated by miRNAs are extremely complex. Each miRNA could have up to hundreds of targets and can be linked, in its action, to other ones; thus, it is necessary to analyze groups of miRNAs. Until now, no data in the context of GBM survival have been obtained. In the present work, our aim was to analyze, in the light of clinical outcomes, the association between the expression levels of a panel of six miRNAs relevant in brain tumorigenesis (i.e., miR-181c, miR-181d, miR-21, miR-195, miR-196b, miR-648), in order to hypothesize their role as an additional marker for predicting the efficacy of chemoradiation as a GBM treatment [20]. According to Kreth and colleagues, who used a bioinformatics-guided experimental approach, this group of miRNAs was shown to be able to downregulate MGMT expression independently of promoter methylation by elongating the 3′-UTR end of the mRNA [21]. In addition, in GBM cell lines, these miRNAs were capable of influencing the response to TMZ independently of the methylation status of the *MGMT* promoter [21].

Moreover, we described the expression pattern and the interplay between specific miRNA pairs, which will help predict survival following TMZ-based therapy.

## 2. Materials and Methods

Our data were obtained from two cohorts collected from two neurosurgical centers in Switzerland and Italy (Service of Neurosurgery of the Neurocenter of Southern Switzerland, EOC, Switzerland and Department of Neurosurgery at Insubria University Hospital, Italy), encompassing the period between 2004 and 2013. This study was conducted in compliance with appropriate protocols, including the current version of the Declaration of Helsinki, the ICH-GCP or ISO EN 14155 [ISO 14155:2020] (as far as applicable), as well as all national legal and regulatory requirements [21]. The data were collected and analyzed only after approval by the Ethics Committees (Cantonal Ethics Committee, Bellinzona, Switzerland) (Ref. CE 3086-2016-01108).

Clinical data included the gender, age, and type of surgery, as well as postoperative outcome and general follow-up until the death of the patients.

Inclusion criteria were age >18 years, presence of GBM *IDH* wild-type (wt) WHO grade IV, therapy with TMZ according to the Stupp scheme (60 Gray radiotherapy and concomitant chemotherapy with TMZ, followed by six cycles of maintenance TMZ), and tissue availability for molecular analyses.

The exclusion criteria were represented by no clear diagnosis of GBM, pediatric patients (<18 years), patients treated employing schemes of treatment outside the Stupp scheme, and those who died due to GBM-independent causes.

For survival analyses, we collected the OS, defined as the time from surgery to the date of death, and the time to progression (TTP), defined as the length of time from the start of treatment to disease progression.

### 2.1. Histological and Molecular Analysis

Experienced pathologists of the Institute of Pathology, EOC, in Locarno (Switzerland), performed the diagnosis of GBM *IDH* wt WHO grade IV. For each sample, we evaluated *MGMT* promoter methylation and MGMT expression via IHC, and we performed the miRNA analysis.

### 2.2. MGMT Promoter Methylation

Genomic DNA was extracted from three 8 μm thick formalin-fixed, paraffin-embedded (FFPE) tumor sections applying automatic extraction (Maxwell, Promega, Madison, WI, USA). Of the DNA, 100 ng was treated via bisulfite treatment using the EZ DNA Methylation-GoldTM kit (Zymo Research, Irvine, CA, USA), and the *MGMT* methylation status was assessed as previously described [19]. Briefly, 100 ng of DNA was treated via bisulfite and analyzed employing PCR pyrosequencing using the MGMT Plus kit (Diatech Pharmacogenetics, Jesi, Italy). A cut-off of 10% was assumed to define the presence of methylation [19].

### 2.3. MGMT Immunohistochemistry

Three 1–2 μm thick FFPE tissue sections were analyzed for MGMT expression at the protein level using IHC. The methodology applied followed the same protocol described in the paper by Cardia et al. [19]. In brief, the deparaffinization of FFPE tissue was followed by rehydration and pretreatment with citrate buffer (pH6). Then, the sections were incubated overnight with primary anti-MGMT, clone MT3.1 (Chemicon International, Temecula, CA, USA) diluted 1/400 [19].

On the basis of relevant studies, we considered MGMT IHC-positive cases those with intense nuclear staining in more than 5% of neoplastic cells [22,23]. IHC slides were evaluated independently by two pathologists.

### 2.4. miRNA Evaluation

The miRNA extraction was performed using the RecoverAll™ Total Nucleic Acid Isolation Kit for FFPE starting from three 10 μm formalin-fixed, paraffin-embedded (FFPE) tumor sections (ThermoFisher Scientific, Waltham, MA, USA). TaqMan^®^ MicroRNA Reverse Transcription Kit was applied for miRNA-specific retrotranscription, in addition to 5X primers included in the TaqMan MicroRNA assays (Life Technologies, Carlsbad, CA, USA) for miR-181c, miR-181d, miR-21, miR-195, miR-196b, miR-648, and RNU6B (i.e., the endogenous control). Three replicates were performed for each sample using Universal Master Mix and assays from TaqMan MicroRNA assays (Life Technologies, Carlsbad, CA, USA). Twelve normal brain samples of patients with cerebral arteriovenous malformations were used as calibrators for setting up the assays. Relative miRNA expression (assessed by comparing miRNA expression with the mean of normal calibrators) was calculated using the DDCt method.

### 2.5. Statistical Analyses

The basic statistical analysis specifications have already been described [19]. All the computations, graphs, and reports were performed using R software version 4.3.1 (2023-06-16) (The R Foundation for Statistical Computing, Vienna, Austria) and “survival” R package version 3.5-7. In the following sections, we report on the results of the survival analysis of groups of patients (in terms of the OS and TTP) characterized by different levels of miRNA. Then, we describe the survival of the groups of patients defined by different configurations of miRNA pairs. For example, one group may be defined by high levels of two miRNAs and the second group by all other configurations defined by cut-offs.

Regarding miRNA expression, on the basis of our previous relevant studies, we defined three different cut-offs for the evaluation of positive cases: Cut-off > 3; Cut-off > 1; Cut-off > median value. We only reported the results obtained when considering a threshold of 3, according to two previous papers from our research group [19,24], which is also the strongest method. 

## 3. Results

### 3.1. Clinical–Pathological Characteristics of the Cohort and Molecular Data

In this study, we retrospectively included, from January 2004 to December 2013, 112 GBM *IDH* wt WHO grade IV patients. The cohort analyzed was the same as that published by Cardia et al. in March 2023 [19]. A table with clinicopathological data is reported in this previous publication.

### 3.2. Molecular Data: MGMT Promoter Methylation, MGMT IHC, Single-miRNA Expression and Their Association with Survival

Molecular data concerning *MGMT* promoter methylation, MGMT IHC, miRNA expression, and the association between *MGMT* and miRNA expression have already been described by our group in the aforementioned paper [19], along with the association between survival (in terms of OS, PFS, TTP) and *MGMT*, and between survival (in terms of OS, PFS or TTP) and single-miRNA expression [19].

### 3.3. Association between miRNAs

From bivariate analyses (Table 1), we were able to define the miRNA pairs of which the expression was significantly associated. The low expression of miR-181c was associated with the low expression of miR-181d, miR-195, and miR-648 (*p* = 0.0005, *p* = 0.0005, and *p* = 0.0145, respectively) and the high expression of mi-R21 and miR-196b (*p* = 0.0009 and *p* = 0.0005, respectively). The same associations were observed for miR-181d (*p* = 0.0005; *p* = 0.0390; *p* = 0.0345; *p* = 0.0025). On the other hand, miR21 was not associated with miR-648 (*p* = 0.2269), but (considering only the associations not described) its high expression correlated with the high levels of miR-195 and high miR196b (*p* = 0.0005 and *p* = 0.0005, retrospectively). Considering only the missing associations, the high miR-195 expression correlated with high miR-196b expression (*p* = 0.0005), whereas its low expression correlated with low miR-648 expression (*p* = 0.0020); miR-196b did not correlate with miR648 (*p* = 0.2859). A representation of the associations between miRNA expressions is reported in Figure 1.

### 3.4. Comparison of miRNA Pairs with the Group of the Other Four miRNAs in Terms of OS and TTP

Due to the fact that, in our cohort, the miRNA pair groups were sometimes represented by low numerosity, we decided to perform statistical evaluations in terms of survival, comparing single-miRNA pairs, with respect to OS and TTP, against the group of all the other miRNA pairs analyzed in this study. We investigated the miRNA pairs according to the expression trends reported in Figure 1 (Section 3.3). The multivariate analysis added to achieve adequate numerosity demonstrated that, regarding the OS, the only miRNA pairs with a statistically significant influence on OS, compared to the other miRNA groups, were the miR-181c (<0.333, low expression) + miR-648 (<0.333, low expression) (*p* = 0.01), and miR-181d (<0.333, low expression) + miR-648 (<0.333, low expression) (*p* = 0.005) pairs. In particular, survival curves show that cases with poorly expressed miR-648 in combination with a low expression of miR-181c or low expression of miR-181d had a worse prognosis than cases characterized by any other poorly expressed miRNA pair. The survival curves of the two statistically significant miRNA pairs are reported in Figure 2. It should be noted that couples with middle-level expression were not considered because these data would not be biologically relevant.

Regarding TTP, none of the miRNA pairs demonstrated statistically significant correlation when compared to the other miRNA groups.

## 4. Discussion

In GBM, the presence of *MGMT* methylation is a predictive factor for response to temozolomide [3,7,8,25]. Despite this, it has been reported that some non-methylated patients respond to therapy. Currently, the explanation for this is a matter of debate and research. As such, the aim of our work was to shed light on miRNA expression in GBM, which is a potential factor that influences clinical response to GBM therapy. Previous research has revealed a group of six miRNAs (miR-181c, miR-181d, miR-21, miR-195, miR-196b, miR-648) relevant in brain tumor-derived cell lines and for brain functions [26,27,28,29,30,31]. In GBM tumorigenesis, these miRNAs have different roles. Five out of six (miR-181c, miR-181d, miR-21, miR-195, miR-196b) can regulate, in addition to many other targets, a common pathway: the PTEN/PI3K/AKT axis [27,28,29,30,31]. When upregulated, miR-21, miR-181c, miR-181d, and miR-196b lead to *PTEN i*nhibition and consequently uncontrolled proliferation. The opposite role can be attributed to miR-195, which, when expressed, can block the translation of PI3K mRNA, leading to decreased cell proliferation. The activity of the miRNAs just described allows for identifying miR-21, miR-181c, miR-181d, and miR-196b with roles similar to an oncogene and miR-195 with a role similar to a tumor suppressor gene (TSG) [27,29,32,33,34]. Regarding miR-648, the only one for which no role in the regulation of the PTEN/PI3K/AKT pathway has been suggested, the target that is more relevant in GBM development is MOBP (myelin-associated oligodendrocyte basic protein) mRNA. The expression of miR-648 leads to myelin production blockage and, as a consequence, cancer [26]. Due to its role, the function of this miR-648 can be considered similar to one of an oncogene.

The six miRNAs described above, much like every miRNA, have a complex nature, and their mechanism of action involves numerous pathways; therefore, they need to be studied in an interconnected manner. Due to the fact that, as far as our knowledge extends, collective data on these miRNAs have not yet been described, we decided to analyze the effect of miRNA pairs in terms of survival in GBM patients. The data that we first obtained refer to the association between one miRNA and another. This evaluation permitted us to demonstrate the reliability of our cohort because most miRNAs that regulate the same pathway correlated with statistically significant results. In particular, miR-181c, miR-181d, and miR-195 were reasonably statistically associated when low-expressed because they are involved in the same common pathway, i.e., PTEN/PI3K/AKT [27,29,32,33,34]. Moreover, the low expression of miR-648, not involved in the PTEN/PI3K/AKT pathway but instead in the regulation of MOBP mRNA [26], was associated with the low expression of miR-181c, miR-181d, and miR-195. Excluding the role of miR-195 as a TSG, this aforementioned association can be interpreted with the fact that the other three miRNAs assume the same role as oncogenes in GBM development. The only miRNA pairs that were not associated were miR-21 + miR-648 and miR-196b + 648; however, miR-21 and miR-196b were associated when overexpressed, and their behavior was comparable: when highly expressed, they associated with the low expression of the other miRNAs, except miR-648. This is explainable by the fact that they regulate the same pathway (i.e., PTEN/PI3K/AKT) [27,29,32,33,34,35].

Once having established the correlations among the most important miRNAs in GBM tumorigenesis, we further investigated the influence of miRNA pairs on survival in terms of OS and TTP. In order to achieve adequate numerosity for statistics, the miRNA pairs comprising significantly associated miRNAs were compared to the group comprising all other miRNA pairs. We excluded all the miRNA pairs statistically significant with middle-expression grade because they had no biological relevance, and, from the multivariate test, we concluded that the only miRNA pairs with a statistically significant influence on OS, compared to the other miRNA groups, were the pairs miR-181c + miR-648 and miR-181d + miR-648 when low-expressed. In particular, cases with poorly expressed miR-648 in combination with a low expression of miR-181c or miR-181d had a worse prognosis compared to cases characterized by any other poorly expressed miRNA pair. Regarding TTP, we did not obtain any statistically significant results.

The comments made above relating to OS are in line with a previous publication focused on MGMT and written by our research group [19]. In fact, in the article by Cardia et al., we showed how miR-181d and miR-648 are associated with unmethylated *MGMT* cases, and miR-181c and miR-648 with MGMT IHC, which, based on literature data, are the features associated with the worst survival. In addition, we reported how miR-648 is associated with OS [19]. As a consequence, we can speculate that the low expression of miR-181c-miR-648 or miR-181d-miR-648 pairs combined with unmethylated *MGMT* or expression of MGMT can predict a poor prognosis linked to a worse response to TMZ. This may seem strange because miR-181c, miR-181d, and miR-648 are oncogenes; however, we can speculate that, in addition to the PTEN/PI3K/AKT pathway, these miRNAs regulate other proteins, not yet known, which contribute to the development and course of GBM. Furthermore, we must keep in mind that the low expression of oncogenes leads to a low proliferation index and, therefore, could result in a worse response to therapies against cell replication. This hypothesis deserves further consideration: to date, at the clinical level, all GBM patients are treated in the same manner without considering the proliferation level. The only prognostic criterion is the possibility of achieving a complete or at least a gross total resection: as we recently demonstrated [6], patients whose tumor is removed macroscopically in its entirety experience the longest survival, followed by patients with a residual tumor characterized by methylation in the promoter of *MGMT*. On the contrary, the proliferation index has, to date, no consequences in terms of treatment or survival. Therefore, our hypothesis needs to be studied in larger cohorts by adding the proliferative index to the other markers routinely used to characterize GBM patients.

One of the strengths of our work is definitely that we are the first to define how the interplay between some relevant miRNA pairs can affect the survival of GBM patients treated with TMZ. In addition, miRNA assessment is more objective than the currently employed method for predicting survival and response to therapy that is based on *MGMT* methylation evaluation. On the contrary, our study also has some limitations. Among them is the small numerosity that does not allow us to obtain statistically significant data for more copies than those mentioned above. Second, the miRNAs that could also be relevant in GBM are many and involve numerous pathways. In the future, we should enlarge the cohort and increase the number of the miRNAs examined.

## 5. Conclusions

To date, as far as we know, no data have ever been reported on the interplay between miRNAs and the effect on the survival of GBM patients treated by TMZ. This work serves as the starting point to outline how determinate miRNA pairs (miR-181c and miR-648; miR-181d and miR-648) can influence survival in cases. Further studies are needed to confirm and extend our findings in larger cohorts.

## Figures and Tables

**Figure 1 cells-13-00276-f001:**
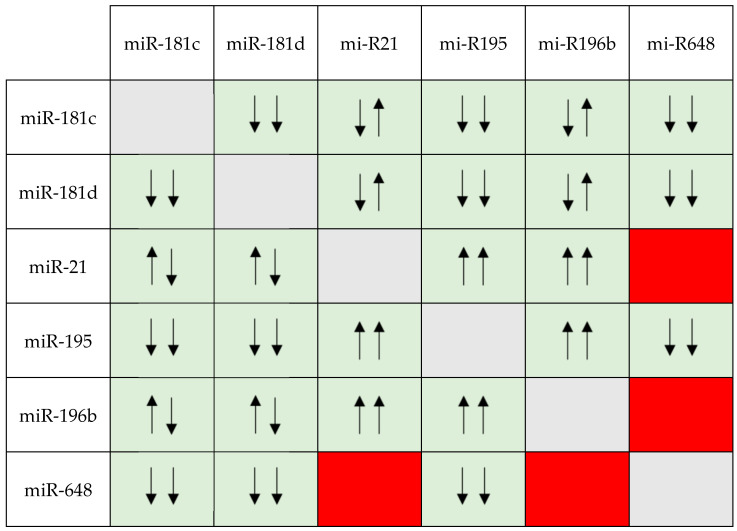
Representation of the associations between miRNA expressions. Green boxes represent the pairs of miRNAs that are significantly associated; red boxes represent no statistically significant association. Up arrows represent the high expression of miRNAs (>3); down arrows represent the low expression of miRNAs (<0.333). The first arrow refers to the miRNA shown vertically, and the second to the miRNA shown horizontally.

**Figure 2 cells-13-00276-f002:**
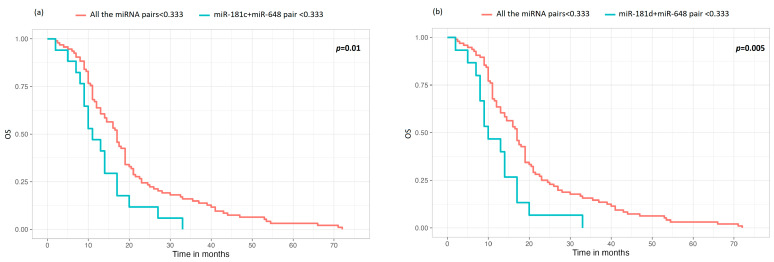
OS curves obtained from the comparison of miRNA pair expression levels that were statistically significant against the group of all the other miRNA pairs analyzed in this study. Only statistically significant data are reported. (**a**) OS miR-181c (<0.333, low expression) + miR-648 (<0.333, low expression) versus low expression of all the other miRNA pairs (*p* = 0.01); (**b**) OS miR-181d (<0.333, low expression) + miR-648 (<0.333, low expression) versus low level of all the other miRNAs (*p* = 0.005).

**Table 1 cells-13-00276-t001:** miRNA bivariate analysis.

		miR-181c			*p*
		<0.333	0.333–3	>3	
miR-181d	<0.333	30/112	8/112	0/112	**0.0005**
		(26.8%)	(7.1%)	(0%)	
	0.333–3	9/112	53/112	9/112	
		(8.0%)	(47.3%)	(8.0%)	
	>3	0/112	1/112	2/112	
		(0%)	(0.9%)	(1.8%)	
miR-21	<0.333	1/112	0/112	0/112	**0.0009**
		(0.9%)	(0%)	(0%)	
	0.333–3	15/112	5/112	0/112	
		(13.4%)	(4.5%)	(0%)	
	>3	23/112	57/112	11/112	
		(20.5%)	(50.9%)	(9.8%)	
miR-195	<0.333	18/112	1/112	0/112	**0.0005**
		(16.1%)	(0.9%)	(0%)	
	0.333–3	21/112	48/112	0/112	
		(18.8%)	(42.9%)	(0%)	
	>3	0/112	13/112	11/112	
		(0%)	(11.6%)	(9.8%)	
miR-196b	<0.333	7/112	0/112	0/112	**0.0005**
		(6.2%)	(0%)	(0%)	
	0.333–3	9/112	1/112	0/112	
		(8.0%)	(0.9%)	(0%)	
	>3	23/112	61/112	11/112	
		(20.5%)	(54.5%)	(9.8%)	
miR-648	<0.333	17/112	11/112	0/112	**0.0145**
		(15.2%)	(9.8%)	(0%)	
	0.333–3	20/112	47/112	10/112	
		(17.9%)	(42.0%)	(8.9%)	
	>3	2/112	4/112	1/112	
		(1.8%)	(3.6%)	(0.9%)	
		**miR-181d**			** *p* **
		**<0.333**	**0.333–3**	**>3**	
miR-21	<0.333	1/112	0/112	0/112	**0.0345**
		(0.9%)	(0%)	(0%)	
	0.333–3	13/112	7/112	0/112	
		(11.6%)	(6.2%)	(0%)	
	>3	24/112	64/112	3/112	
		(21.4%)	(57.1%)	(2.7%)	
miR-195	<0.333	16/112	3/112	0/112	**0.0005**
		(14.3%)	(2.7%)	(0%)	
	0.333–3	21/112	47/112	1/112	
		(18.7%)	(42.0%)	(0.9%)	
	>3	1/112	21/112	2/112	
		(0.9%)	(18.7%)	(1.8%)	
miR-196b	<0.333	6/112	1/112	0/112	**0.0025**
		(5.4%)	(0.9%)	(0%)	
	0.333–3	9/112	1/112	0/112	
		(8.0%)	(0.9%)	(0%)	
	>3	23/112	69/112	3/112	
		(20.54%)	(61.6%)	(2.7%)	
miR-648	<0.333	15/112	13/112	0/112	**0.0390**
		(13.4%)	(11.6%)	(0%)	
	0.333–3	21/112	54/112	2/112	
		(18.7%)	(48.2%)	(1.8%)	
	>3	2/112	4/112	1/112	
		(1.8%)	(3.6%)	(0.9%)	
		**miR-21**			** *p* **
		**<0.333**	**0.333–3**	**>3**	
miR-195	<0.333	1/112	9/112	9/112	**0.0005**
		(0.9%)	(8.0%)	(8.0%)	
	0.333–3	0/112	11/112	58/112	
		(0%)	(9.8%)	(51.8%)	
	>3	0/112	0/112	24/112	
		(0%)	(0%)	(21.4%)	
miR-196b	<0.333	1/112	5/112	1/112	**0.0005**
		(0.9%)	(4.5%)	(0.9%)	
	0.333–3	0/112	3/112	7/112	
		(0%)	(2.7%)	(6.2%)	
	>3	0/112	12/112	83/112	
		(0%)	(10.7%)	(74.1%)	
miR-648	<0.333	1/112	6/112	21/112	0.2269
		(0.9%)	(5.4%)	(18.7%)	
	0.333–3	0/112	14/112	63/112	
		(0%)	(12.5%)	(56.2%)	
	>3	0/112	0/112	7/112	
		(0%)	(0%)	(6.2%)	
		**miR-195**			** *p* **
		**<0.333**	**0.333–3**	**>3**	
miR-196b	<0.333	6/112	1/112	0/112	**0.0005**
		(5.4%)	(0.9%)	(0%)	
	0.333–3	5/112	5/112	0/112	
		(4.5%)	(4.5%)	(0%)	
	>3	8/112	63/112	24/112	
		(7.1%)	(56.2%)	(21.4%)	
miR-648	<0.333	11/112	17/112	0/112	**0.0020**
		(9.8%)	(15.2%)	(0%)	
	0.333–3	7/112	48/112	22/112	
		(6.2%)	(42.9%)	(19.6%)	
	>3	1/112	4/112	2/112	
		(0.9%)	(3.6%)	(1.8%)	
		**miR-196b**			** *p* **
		**<0.333**	**0.333–3**	**>3**	
miR-648	<0.333	4/112	3/112	21/112	0.2859
		(3.6%)	(2.7%)	(18.7%)	
	0.333–3	3/112	6/112	68/112	
		(2.7%)	(5.4%)	(60.7%)	
	>3	0/112	1/112	6/112	
		(0%)	(0.9%)	(5.36%)	

*p*-values obtained from bivariate analysis between miRNA expressions based on the cut-off > 3. Level of significance: *p* < 0.05 (in bold). Abbreviations: *p*, *p*-value.

## Data Availability

The datasets used and analyzed during the current study are available from the corresponding author upon reasonable request. The data are not publicly available due to institutional policy.

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
