# Peer review of "Clinical Relevance and Interplay between miRNAs in Influencing Glioblastoma Multiforme Prognosis"

_cells, 2024, doi:10.3390/cells13030276_

Round 1
Reviewer 1 Report
Comments and Suggestions for Authors
In the review article titled: „ Clinical relevance and interplay of miRNAs in influencing GBM prognosis” Epistolio S and colleagues described the role certain microRNAs in the development, aggressiveness and drug resistance in glioblastoma multiforme. The title and content of the article raise an extremely important scientific problem, which is a development of glioblastoma and the problems with its management. Glioblastoma multiforme is an extremely malignant and heterogenous cancer with a very poor outcome. Since decades the mechanism of its survival has been so weakly understood that no effective treatment has been proposed until now. Hence there is an urgent need to solve the mystery of its survival, what helps in development of its efficient management. The authors of the article focus their attention on the importance of certain miRNAs in development and survival of GBM in relation to MGMT expression and outcome. Since the significance of particular miRNAs in pathogenesis of diverse solid tumors was confirmed, the choice of the topic is highly justified. Additionally, regarding the chapter titled Materials and Methods, it is easy to recognize that the authors conducted scientific research thoughtfully. First, they collected a sufficient number of samples for further evaluation and next they properly assessed the expression of MGMT and certain miRNAs. The authors also did not try to stretch the results, introducing only the obtained results, not the probable ones. Moreover, the way of presenting the line of reasoning and the quality of English language are very interesting.
However, I found some information in the text that can make a readers to become confused. In the manuscript the authors significantly emphasized that miR-21, miR-181c, miR-181d, miR-196b functions as an oncogenes in GBM, what could easy suggest that reducing their expression could have a beneficial impact on the clinical outcome. However, authors interpreting the results of their own research demonstrate that low expression of miR-181c and miR-181d correspond to poor overall survival. Both these opposing facts wouldn't be surprising, if only authors better explain that situation. One possible explanation could be that miR-181c and miR-181d may control the function of yet unknown single proteins or signaling pathways (except the ones we know- described in the text PTEN/PI3K/AKT signaling pathway) that contribute to the development and/or aggressiveness of GBM. The statement that low expression of oncogenes corresponds to low proliferation index and in consequence to worst response to the treatment focused to cell replication is only partially important from clinical point of view. In case of tumors with low proliferation (proliferative) index we can expect that surgery, radiation or targeted treatment may be successful. In these circumstances I would like to ask to develop this topic in a more comprehensive way in the discussion.
Reviewer 2 Report
Comments and Suggestions for Authors
The original article "Clinical relevance and interplay of miRNAs in influencing GBM prognosis", presented by Samantha Epistolio et al., is a description of some statistical studies of some miRNAs analyzed in 112 GBM patients.
I have minor and a major task to comment.
Minor tasks:
I miss some more references to previous studies and meta-analysis developed in GBM such us:
- Expression of 19 microRNAs in glioblastoma and comparison with other brain neoplasia of grades I-III (PMID: 24412053 PMCID: PMC5528554 DOI: 10.1016/j.molonc.2013.12.010)
and
- Potential role of microRNAs as biomarkers in human glioblastoma: a mini systematic review from 2015 to 2020 (PMID: 34032976 DOI: 10.1007/s11033-021-06423-9).
I think these studies, among others in general carcinomas, would put some light on the miRNAs selected.
Related to this, most of the data are based on a previous article recently published and, therefore, few information is provided about the reason for selecting these miRNAs. I think readers need deeper explanation rather than a reference.
The way the table and analysis is explained is not clear.
Major task:
If the ground of the manuscript is to find a reason about of that 20% of unmethylated MGMT gene, why the statistics and the survival curves are not individually analyzed between these two big groups (methylated and unmethylated).
I think, analysis like this would provide much more information.
Comments on the Quality of English Language
English is fine, from my point of view. I think only minor changes can be done.
Round 2
Reviewer 2 Report
Comments and Suggestions for Authors
Authors have answered all the comments and questions indicated. Therefore, I approve the publications of this manuscript.
Author Response
Dear reviewer,
thank you for you revisions and suggestions.
On behalf of all authors, I send my best regards.
Samantha Epistolio